# Rapid Absorption of Naloxone from Eye Drops

**DOI:** 10.3390/ph15050532

**Published:** 2022-04-25

**Authors:** Johanna Tuunainen, Lasse Saloranta, Jouko Levijoki, Jenni Lindstedt, Jenni Lehtisalo, Sari Pappinen, Meri Ramela, Sami Virtanen, Heikki Joensuu

**Affiliations:** 1Orion Corporation Orion Pharma, Orionintie 1A, FIN-02200 Espoo, Finland; lasse.saloranta@orionpharma.com (L.S.); jouko.levijoki@orionpharma.com (J.L.); jenni.lindstedt@orionpharma.com (J.L.); jenni.lehtisalo@orionpharma.com (J.L.); sari.pappinen@orionpharma.com (S.P.); meri.ramela@orionpharma.com (M.R.); sami.virtanen@orionpharma.com (S.V.); 2Department of Oncology, Helsinki University Hospital, Haartmaninkatu 8, FIN-00290 Helsinki, Finland; heikki.joensuu@helsinki.fi

**Keywords:** naloxone, ocular administration, opiate, pharmacokinetics, tolerability

## Abstract

Naloxone as emergency treatment for opioid overdosing can be administered via several routes. However, the available administration methods are invasive or may be associated with incomplete or slow naloxone absorption. We evaluated pharmacokinetics and local tolerance of naloxone ocular drops in healthy beagle dogs. Naloxone administration as eye drops produced fast absorption with time to maximum plasma concentration (t_max_) achieved in 14 to 28 min, high plasma exposure (C_max_ 10.3 ng/mL to 12.7 ng/mL), and good bioavailability (41% to 56%). No signs of ocular irritability were observed in the scored ocular tolerability parameters, and the reactions of dogs suggesting immediate ocular discomfort after the dosing were sporadic and short lasting. Slight and transient increase in the intraocular pressure and transient decrease in the tear production were recorded. The results suggest that eye drops may provide a fast and an effective non-invasive route for naloxone administration to reverse opioid overdosing, and clinical studies in the human are warranted.

## 1. Introduction

Opioid overdose associated events, such as respiratory depression and cardiac arrest that may lead to hospitalization and death, continue to increase in some countries including the USA, where opioids caused 49,865 deaths in 2019 [1,2]. Of these deaths, 72.9% were due to synthetic opioids. Nonfatal opioid overdose events far exceed the numbers of opioid-related deaths. The U.S. Centers of Disease Control and Prevention (CDC) Drug Overdose Surveillance and Epidemiology (DOSE) registry reported a total of over 20,000 emergency department visits each month due to suspected opioid overdose in 42 U.S. states during May through September 2020 [3].

Acute opioid overdose may cause a triad of coma, miosis, and respiratory depression, requiring immediate treatment. Naloxone, a competitive opioid antagonist that reverses effectively opioid toxicity, is a critical part of the first aid. However, in the era of synthetic opioids, higher and more frequent naloxone doses may be needed [4]. To prevent opioid overdose related deaths, both the U.S. Food and Drug Administration (FDA) and the European Medicines Agency (EMA) promote availability of naloxone [5,6]. Take-home naloxone programs aiming at training of individuals who are at risk of experiencing or witnessing opioid overdose are an important strategy in avoiding overdose deaths [7].

Besides parenteral administration, naloxone can be administered sublingually, buccally, intranasally, and via the respiratory tract. A limitation for the sublingual, buccal, and nasal administration routes is impaired absorption caused by vomiting or secretions. The respiratory tract route is not preferable when opioid overdosing causes respiratory depression [8].

We hypothesized that administration of naloxone via the ocular route as eye drops might offer a rapid means to treat opioid overdosing by health care professionals or even by non-professionals as an emergency procedure, provided that high enough naloxone plasma concentrations can be achieved rapidly. Drug delivery to the systemic circulation using eye drops has rarely been used clinically on purpose. Reasons for this include possible ocular irritation and other local side effects, especially in the long-term continuous treatment, and the variability in absorption due to spillage on the cheek. In addition, the small-dose volume sets limit the dose that can be administered and requirements to the physico-chemical properties and potency of the drug molecule. Yet, ocular administration is an interesting option, because nearly complete bioavailability and rapid absorption may be achieved with relatively small lipophilic drugs [9]. Naloxone fulfils this requirement since it has a molar mass of 327 g/mol and logP of about two [10]. The conjunctiva of the eye and the nasal mucosa are the main sites of systemic drug absorption from eye drops. Both tissues are highly vascularized, favoring rapid drug absorption [9].

The aim of the current study was to assess the plasma pharmacokinetics (PK) and local tolerance of naloxone administered as eye drops to healthy dogs. The results show that therapeutic naloxone plasma concentrations can be achieved easily and rapidly in the dog with eye drops. To the best of our knowledge, this is the first study that has evaluated naloxone administration as eye drops.

## 2. Results

### 2.1. Pharmacokinetics

In the first PK study, naloxone plasma concentration increased rapidly following ocular administration (Figure 1). With the highest dosing used (3 × 100 µg/kg), naloxone plasma concentration of 10 ng/mL was achieved in about 10 min. The plasma exposure of naloxone increased dose-dependently, and the maximum naloxone plasma concentrations (C_max_) was reached in 19 to 23 min, regardless of the naloxone dose used (Table 1). When naloxone was administered as eye drops at the dose of 3 × 100 µg/kg, a similar naloxone plasma exposure (area under the plasma concentration-time curve, AUC) was achieved as with the intravenous 100 µg/kg dose based on descriptive statistics. The bioavailability of naloxone from eye drops was 37%, 41%, and 29% with the dose of 1 × 20 µg/kg, 1 × 100 µg/kg, and 3 × 100 µg/kg, respectively.

In the second PK study, the C_max_ of naloxone was achieved in 14 to 28 min regardless of the eye drop formulation used (Figure 2). The highest maximum (C_max_) and total plasma exposure (AUC_last_) of naloxone were observed with a citrate buffer vehicle. The bioavailability of naloxone from eye drop formulations was 45% to 56% (Table 2).

### 2.2. Clinical Signs and Local Tolerability

No signs of ill health, and no change in the appearance or behavior were recorded during the observation periods in the first PK study. In the second PK study, mild transient tremor was observed after administration of each of the ocular formulations in one or two dogs out of the six dogs tested at either the 1 h time point or the 1.5 h time point. In addition, one dog vomited 1.5 h after the dosing of the formulation 2.

We detected no conjunctival discharge, swelling, hyperemia, blepharospasms, or protrusion of the third eyelid with any of the treatment groups. The reactions captured during the 30 min intensive observation period that immediately followed the dosing consisted of transient squinting of the eyes in 2–4 dogs out of the six dogs per group and occasional pawing of the ocular area (a maximum of 2 dogs per group). All reactions were graded as mild. We found no significant differences in these parameters between the different eye drop formulations used.

Naloxone induced a slight transient increase in the intraocular pressure (IOP). The mean IOP values during all 46 treatment periods were 19.7 ± 2.64, 22.2 ± 3.00, and 20.2 ± 2.15 mmHg at pre-dosing, and 2 h and 6 h after the dosing, respectively (Figure 3, upper panel). There was no significant difference (*p* = 0.22) between the ocular or intravenous dosing routes in the change from the baseline IOP (Appendix A).

Naloxone induced a transient decrease in the tear production. The Schirmer’s tear test (STT) mean values for all 46 treatment periods were 17.5 ± 2.87 mm/minute, 14.0 ± 3.77 mm/minute, and 15.0 ± 3.33 mm/minute at pre-dose, and 2 h and 6 h after naloxone administration, respectively (Figure 3, lower panel). There were no significant differences in the magnitude of the tear production change from the baseline value between the different naloxone formulations or between the intravenous and the ocular dosing routes (*p* = 0.07; Appendix A), except that the ocular formulation 3 (propylene glycol plus ethylenediaminetetraacetic acid disodium salt (EDTA) in citrate buffer) caused a larger decrease in the tear production compared to intravenous naloxone administration 6 h after administration (*p* = 0.02). The lowest recorded individual STT value was 7 mm/minute.

## 3. Discussion

We describe a novel route for naloxone administration, the ocular route as eye drops. Ocular administration resulted in a dose-dependent increase in the plasma naloxone concentration when the naloxone concentration in the eye drop was increased from 3 mg/mL to 15 mg/mL. Importantly, the naloxone plasma levels increased rapidly after ocular administration, and the dogs tolerated the eye drops well. Solubility of naloxone in the solvents used was high, which enabled increased absorption with higher concentrations of naloxone in the eye drops.

To our knowledge, there are no data available from naloxone administration as eye drops to humans. The present findings from dogs suggest that ocular administration of naloxone might be a useful and practical method to address the increasing death toll from opioid misuse. Opioid use-related overdoses and overdose-caused deaths have increased steadily during the past 10 years; therefore, the FDA called a public meeting for new options to increase the availability of naloxone [11]. As part of the initiative, the FDA approved naloxone formulations freely available from pharmacies as over the counter (OTC) products [12]. Alternative naloxone administration routes to parenteral administration, i.e., sublingual, buccal, nasal, and inhalation administration have been developed for use at home or at public areas, but these administration routes have potential limitations, such as invasiveness, impaired naloxone absorption, or decreased naloxone access to the target tissues due to opioid overdose-related respiratory depression [8]. Eye drop administration is a non-invasive procedure that does not require any special apparatus or training, and could, therefore, be well-suited for emergency use, even by non-professionals.

All tested naloxone formulations were well tolerated. Few signs of ocular irritability were detected, and the reactions of dogs suggesting immediate ocular discomfort after the dosing were sporadic and short-lasting. The ocular naloxone formulations induced small and transient increases in the IOP and a transient decrease in the tear production. However, similar adverse effects were also observed with intravenously administered naloxone. Furthermore, most of the measured Schirmer’s tear test and IOP values remained within the normal physiological range, and only sporadic recorded values were outside of the reference range [13,14]. Therefore, the effects of naloxone on the IOP and tear production can be considered clinically insignificant considering a single treatment.

The study has some limitations. Ocular administration of naloxone was studied in beagle dogs only, and the number of dogs tested was limited. The beagle dog is considered a good model for studying ocular drug administration [15,16]. A crossover design was used to increase the statistical power, and the results were consistent despite the limited number of dogs. We did not compare ocular administration with intranasal or intramuscular administration, or with novel experimental devices such as hollow microneedles [17]. This remains an important topic for further research. However, the naloxone eye drop dose of 100 µg/kg may compare well with a 4 mg (~170 µg/kg) intranasal naloxone dose, since when this intranasal dose was administered to dogs using a commercially available single-dose naloxone atomizer developed for human intranasal administration, the achieved naloxone plasma exposure (mean C_max_) was lower than in the present study with 100 µg/kg eye drops (9.3 ng/mL vs. 12.2 ng/mL, respectively). The bioavailability was also lower (32% vs. 41%) and the absorption time (T_max_) was roughly similar (21.6 vs. 22.5 min) [18]. In humans, intranasal and intramuscular administration resulted in equally fast absorption of naloxone [12,19,20], suggesting that absorption of naloxone from eye drops might compare well to intramuscular administration. We lack human data, but we hope that the current findings in dogs will encourage studying of ocular naloxone administration in a registered clinical trial.

## 4. Materials and Methods

### 4.1. Animals

The animal studies were performed according to the rules of the Council of Europe and the National Research Council, U.S.A.9–12 All study procedures were approved by the National Animal Experiment Board of Finland, license number ESAVI/21320/2018.

Laboratory beagle dogs were used in the study; in the first PK study, two females and four males (weight, 7 kg to 13 kg) were used, and in the second PK study, six males (weight, 8 kg to 12 kg) were used. The dogs were purchased from Marshall BioResources (North Rose, New York, NY, USA, for the first PK study; or Gannat, France, for the second PK study) and housed at Orion Pharma, Turku, Finland. The dogs were housed in groups of two to three dogs in solid floor pens (2.0 × 3.8 m) equipped with resting shelves. During the naloxone dosing days, the dogs were housed individually in the pens, so that the dogs had a visual contact to each other. The dogs were monitored daily by the laboratory personnel. The room temperature was maintained at 19 ± 2 °C and humidity at 55 ± 15%. In the light–dark cycle, the lights were kept on from 06.00 to 18.00 h. The dogs had free access to tap water from a public supplier and they were offered pelleted dog feed (LabDiet^®^ 5L66, PMI Nutrition International, Richmond, IN, USA) approximately 200 g day/dog at 11 a.m.

The dogs were fasted overnight and for at least 12 h before naloxone dosing. The body weight was measured 1 to 2 days prior to dosing. On the naloxone dosing days, food was offered approximately four hours after the dosing.

### 4.2. Chemicals

Naloxone HCl was purchased from Enzo Life Sciences (Lausen, Switzerland). Propyleneglycol (Ph.Eur, USP) was purchased from Merck (Darmstadts, Germany) and EDTA from Merck (Barcelona, Spain), citric acid monohydarate and trisodium citrate dihydrate (Ph.Eur, USP) were purchased from Jungbunzlauer (Wulzeshofen, Austria).

### 4.3. The First Pharmacokinetic Study—Dose Finding

The first PK study was conducted to investigate the preliminary PK, the dosing, and the local tolerance of naloxone administration as eye drops. The dose finding was based on the published human data about naloxone as a rescue medication for opioid overdosing [19,20]. The naloxone eye drop doses were selected to produce similar plasma concentrations in dogs as were achieved in humans with FDA-approved intranasal or intramuscular devices that can be used by non-professionals for administration of naloxone. In these studies, the C_max_ of naloxone in plasma after a 2 mg to 8 mg intranasal dose and a 0.4 mg to 2 mg intramuscular dose ranged from 2.8 to 10.9 ng/mL, and from 0.8 to 4.7 ng/mL, respectively [19,20].

Ascending doses of naloxone were administered until the desired plasma concentration level was reached. The starting dose was 20 µg/kg, and the dose was subsequently increased to 100 µg/kg. After reaching the 100 µg/kg dosing level, the 100 µg/kg dosing was repeated three times (3 × 100 µg/kg) to assess response and tolerability to repeated ocular dosing. With the single 20 µg/kg and 100 µg/kg doses, the formulation concentration of 3.0 mg/mL and 15.0 mg/mL, respectively, were administered in a volume of 6.7 µL/kg, and with the repeated dosing (3 × 100 µg/kg), the 6.7 µL/kg volume was administered three times at 2 min intervals.

Eye drop formulations were prepared by dissolving naloxone in 20 mM citrate buffer. The pH of the eye drop solutions was adjusted to 4.0 (3.95–4.04) with 1M NaOH or 1M HCl. The solutions were sterile filtrated immediately before use. For intravenous dosing, 0.3 mg/mL of naloxone was given over 30 s in saline (0.9% NaCl solution) using a dosing volume of 335 µL/kg. Six dogs received naloxone ocularly and four dogs intravenously. The dogs were crossed over between the naloxone dosing groups, and the washout period between the treatment periods was at least seven days.

A blood sample to measure the naloxone plasma concentration was drawn just before the dosing, and 3, 5, 10, 20, 30, and 45 min, and 1, 2, 4, 6, and 8 h after the start of the dosing, respectively. In addition, a blood sample was drawn one minute after the start of the 3 × 100 µg/kg dosing. The 8 h blood sample was not collected in the 20 µg/kg dosing group.

### 4.4. The Second Pharmacokinetic Study—Formulation Comparison

The second PK study focused on unblinded comparisons of efficacy and safety of three eye drop formulations. In each of the three formulations the naloxone concentration was 15.0 mg/mL and the dosing volume was 6.7 µL/kg, resulting in a naloxone dose of 100 µg/kg.

Formulation 1 was prepared by dissolving naloxone in 20 mM citrate buffer. Formulation 2 was obtained by dissolving naloxone in a solution containing 10% propylene glycol and 0.3% EDTA in sterile water, and formulation 3 by dissolving naloxone in a solution containing 10% propylene glycol and 0.3% EDTA in 20 mM citrate buffer. The pH of the eye drop solutions was adjusted to 4.0 (3.95–4.04) with 1M NaOH or 1M HCl. The solutions were sterile filtrated immediately before use.

In addition to ocular administration to six dogs, naloxone was also administered intravenously to six dogs, to compare the systemic bioavailability of naloxone from eye drops to naloxone intravenous administration. For intravenous administration, naloxone was dissolved in saline (0.9% NaCl solution) at a concentration of 0.3 mg/mL and given at the dosing volume of 335 µL/kg over 30 s.

A blood sample to measure naloxone concentration was drawn just before the dosing and 2, 5, 10, 15, 30, and 45 min, and 1, 1.5, 2, 4, 6, and 24 h after the start of the dosing, respectively. The dogs were crossed over between the dosing groups, and the washout period between the treatment periods was at least 7 days.

### 4.5. Drug Administration and Blood Sampling

A precise volume of the formulation was dropped on the corneal surfaces of the eyes using a manual single channel pipette (Finnpipette, Thermo Fisher, Scientific, Vantaa, Finland) while an assisting person held the dog´s head gently but firmly in a stable position. Signs of spillage or overflood were carefully monitored during and after the administration. In intravenous administration, a slow (30 s) intravenous bolus was administered via a cannula inserted in the cephalic vein.

Blood samples (~1 mL) were taken into pre-cooled K2-EDTA tubes from the cephalic vein (after eye drop dosing) and the jugular vein (all pre-dose samples and after intravenous dosing). The sampling times were recorded. The blood samples were kept on ice for a maximum of 30 min, followed by the separation of plasma by centrifuging the tubes at 1700–1800× *g* for 10 min at +4 °C. The plasma was divided into two aliquots which were frozen immediately and stored in polypropylene tubes in an upright position at −80 °C.

### 4.6. Naloxone Plasma Concentration Analysis

Naloxone concentration in the dog plasma was measured using a liquid chromatography tandem mass spectrometry (LC-MS/MS) method. The plasma samples were prepared for the analysis using Waters Sirocco protein precipitation plates (Milford, USA). An aliquot of 30 µL of the plasma sample was loaded to the plate and precipitated with 100 µL of acetonitrile. The loaded plate was vortex-mixed, and the supernatant was separated by centrifugation at 3020× *g* for 10 min and concentrated to dryness. The evaporated residue was dissolved in 100 µL of mobile phase consisting of 0.1% formic acid and acetonitrile. Chromatographic separation was carried out on a Phenomenex Synergy polar RP (Torrance, CA, USA) or a Waters Acquity HSS T3 column (Wexford, Ireland). The samples were analyzed with a triple quadrupole mass spectrometer (AB Sciex API 4000, Singapore) coupled with a Waters Acquity ultra-performance liquid chromatography (Milford, MA, USA) or a Shimadzu UPLC (Koyoto, Japan). The analyte was detected with multiple reaction monitoring and quantified using external standardization. The lower limit of quantification for naloxone in the plasma ranged from 0.05 ng/mL to 0.5 ng/mL. Quality control samples at four concentration levels were prepared and analyzed in duplicate among the study samples in each sample batch. The method was acceptably performed in terms of specificity, sensitivity, calibration range, precision, and accuracy.

### 4.7. Clinical Signs and Ocular Tolerability

The dogs were observed frequently on the dosing days, and signs of ill health, reactions to the treatment, and changes in the appearance or behaviors (such as postural changes or dysphoria) were recorded. Ocular reactions to naloxone administration were monitored continuously for the first 30 min. Signs suggesting ocular discomfort or irritation were rated at pre-dose, and 30, 60, 120 min (both PK studies), and at 240 min (the second PK study) after eye drop administration, respectively. The predetermined parameters suggesting local intolerability included conjunctival hyperaemia, swelling, and discharge as signs of local tissue irritation, and blepharospasm, protrusion of the third eyelid, and itching as signs of ocular discomfort. Each sign was rated on a 4-point scale from 0 (none) to 3 (severe). On the non-dosing days, the dogs were observed at least once daily, and any abnormal findings were recorded.

A STT, performed by keeping a filter paper (Optitech Eyecare, Allahabad, India) tip inside the lower eyelid for 60 s, was done to measure tear production before the treatment (baseline), and 2 h and 6 h after naloxone dosing (the second PK study). The IOP was measured using rebound tonometry (icare Tonovet Plus, Vantaa, Finland) at the same timepoints as tear production.

### 4.8. Statistical Analysis

Statistical analyses were carried out using the SAS software (SAS^®^, SAS Institute Inc., Cary, NC, USA). Analysis of variance (ANOVA) with 2-sided 0.05 significance level with 95% confidence intervals was applied to the log-transformed PK parameters (AUC_last_ and C_max_) of the second PK study. The effects of naloxone on the IOP and tear secretion were analyzed with repeated measures analysis of covariance (RM ANCOVA). The model incorporated terms for the treatment, time, and the treatment * time interaction. Both the within subject variation (each subject contributed data from two eyes) and repeated measurements during the treatment were modelled. The statistical analyses were performed for the observed cases only and no adjustments for multiplicity were made. A 2-sided 0.05 significance level with 95% confidence intervals was used for all effects of interest in the models. Model-based treatment difference estimates were calculated. The normality assumption was examined primarily by visual inspection of the residuals for the continuous variables, e.g., by using the normal probability plot.

## 5. Conclusions

Naloxone reached the systemic circulation rapidly from eye drops. The current results suggest that eye drops may provide a fast and effective non-invasive method to administer naloxone for reversing opioid overdosing. Eye drops are quick and easy to administer and do not require costly administration devices. The studied naloxone eye drops did not cause ocular discomfort in the dog. Transient alterations in the intraocular pressure and the tear production may have limited clinical significance in emergency situations. The present findings warrant further investigation of naloxone eye drops in humans, since they could potentially become a novel and practical emergency treatment for opioid overdosing.

## Figures and Tables

**Figure 1 pharmaceuticals-15-00532-f001:**
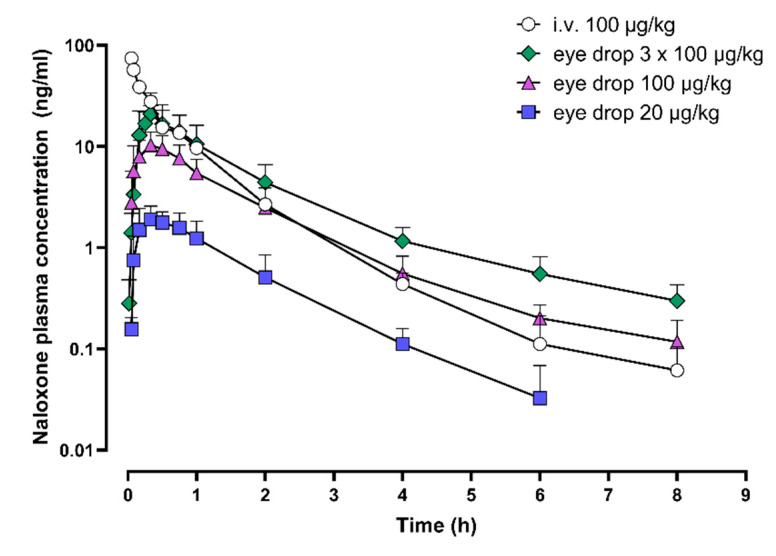
Naloxone plasma concentrations after intravenous administration and after ocular administration. Upper panel: naloxone plasma concentrations during the first 8 h after administration; lower panel: during the first hour after administration. Mean + standard deviation are shown. Four dogs received naloxone intravenously and six as eye drops. i.v., intravenous.

**Figure 2 pharmaceuticals-15-00532-f002:**
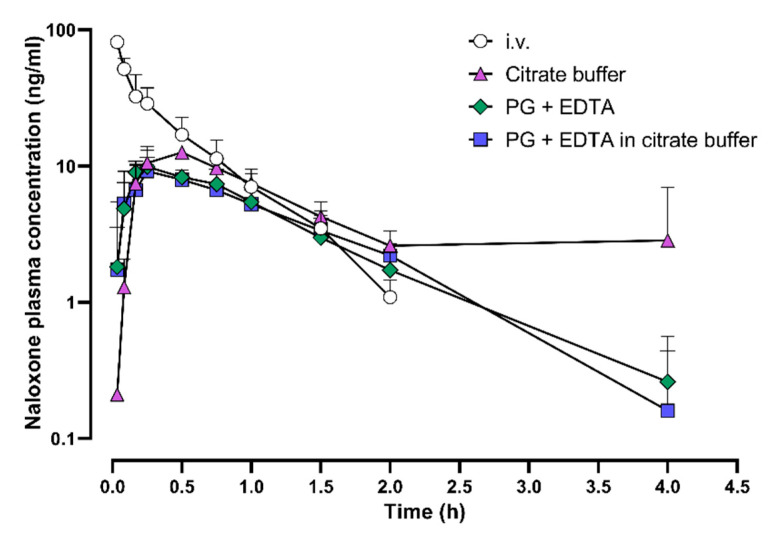
Naloxone plasma concentrations after intravenous administration and after ocular administration using three different formulations. Upper panel: naloxone plasma concentrations during the first 4 h after administration; lower panel: during the first hour after administration. Mean + standard deviation are shown. Six dogs received naloxone intravenously and six dogs received each of the eye drop formulations. EDTA, ethylenediaminetetraacetic acid; i.v., intravenous; PG, propylene glycol.

**Figure 3 pharmaceuticals-15-00532-f003:**
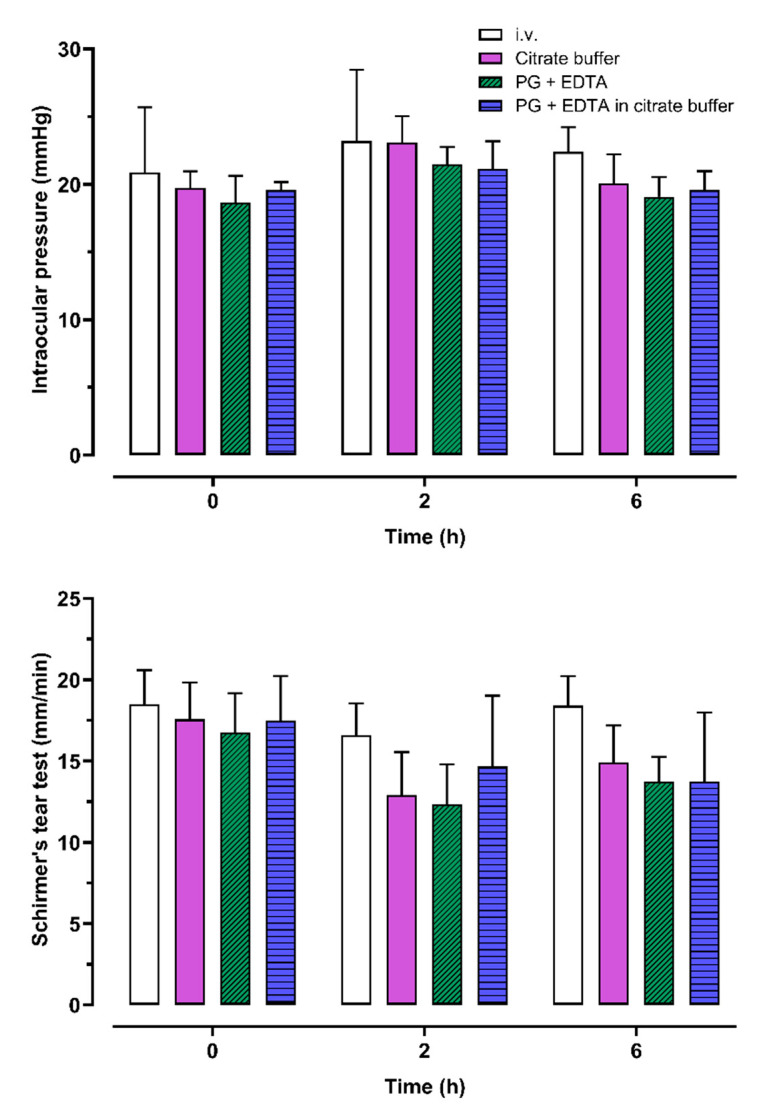
Intraocular pressure (upper panel) and tear production (lower panel) at baseline, 2 h, and 6 h, after either intravenous or ocular administration of 100 µg/kg naloxone to six male dogs. EDTA, ethylenediaminetetraacetic acid; i.v., intravenous; PG, propylene glycol.

**Table 1 pharmaceuticals-15-00532-t001:** Pharmacokinetic parameters of naloxone after intravenous and ocular administration.

Dosing Route	Dose		C_max_ ng/mL	t_max_ h	AUC_last_ h × ng/mL	t_last_ h	BA %
	µg/kg	µL/kg					
Intravenous	100	6.7			33.5 ± 5.3	7.0 ± 2.0	
Ocular	20	6.7	2.2 ± 0.5	0.38 ± 0.22	3.1 ± 1.2	5.0 ± 1.1	37 ± 10
Ocular	100	6.7	12.2 ± 2.9	0.36 ± 0.24	15.4 ± 5.0	7.7 ± 0.8	41 ± 10
Ocular	3 × 100	3 × 6.7	23.5 ± 4.6	0.32 ± 0.11	28.4 ± 11.0	8.0 ± 0.0	29 ± 10

Notes: Mean ± standard deviation are shown. Four dogs received naloxone intravenously and six dogs ocularly. Abbreviations: AUC_last_, area under the plasma concentration–time curve from administration to the last measurable plasma concentration; BA, bioavailability; C_max_, maximum plasma concentration; t_last_, time of last measurable plasma concentration; t_max_, time to C_max._

**Table 2 pharmaceuticals-15-00532-t002:** Pharmacokinetic parameters of naloxone after ocular administration in three vehicles.

Dosing Route	Vehicle	Dose		C_max_ ng/mL	t_max_ h	AUC_last_ h × ng/mL	t_last_ h	BA %
		µg/kg	µL/kg					
Intravenous	Saline	100	6.7			27.9 ± 7.5	2.0 ± 0.0	
Ocular	Citrate buffer	100	6.7	12.7 ± 1.3	0.46 ± 0.10	16.5 ± 5.3	3.0 ± 1.1	56 ± 14
Ocular	PG + EDTA	100	6.7	10.3 * ± 1.8	0.28 ± 0.11	11.3 * ± 3.0	2.7 ± 1.0	45 ± 10
Ocular	PG + EDTA in citrate buffer	100	6.7	11.2 ± 2.4	0.23 ± 0.16	10.9 * ± 2.4	2.0 ± 0.8	48 ± 5.0

Notes: Mean ± standard deviation are shown. Six dogs received naloxone intravenously and ocularly. Abbreviations: AUC_last_, area under the plasma concentration–time curve from administration to the last measurable plasma concentration; BA, bioavailability; C_max_, maximum plasma concentration; EDTA, ethylenediaminetetraacetic acid; PG, propylene glycol; t_last_, time of last measurable plasma concentration; t_max_, time to C_max._ * *p* < 0.05 compared to the citrate buffer vehicle.

## Data Availability

The study data are available from the corresponding author upon reasonable request.

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
