# Peer review of "Rapid Absorption of Naloxone from Eye Drops"

_pharmaceuticals, 2022, doi:10.3390/ph15050532_

Round 1

Reviewer 1 Report

The manuscript “Rapid absorption of naloxone from eye drops” by Tuunainen et al reported evaluation of the pharmacokinetics and local tolerance of naloxone ocular drops in healthy beagle dogs. The manuscript is well-written and the work shows promise in terms of providing a non-invasive, rapid, and effective way to administer naloxone to reverse opioid overdosing via eye drops. Yet, authors must address the following issues:

- The most concern in this work is that authors ignored the precorneal losses and the barrier function of the cornea. Literature for ophthalmic administration reported that for effective systemic delivery a relatively high drug concentration needs to be circulating in the blood plasma in order to achieve a therapeutically effective dose within the eye.

- The introduction section needs more information about the ocular drug delivery especially the obstacles encountered during ophthalmic administration.

- Lines 188-192 need rephrasing to make it clear for the reader.

- The conclusion section needs to be re-written. Authors should comments on their finding and the advantages of the obtained results.

Author Response

Response to Reviewer 1 Comments

Point 1: The most concern in this work is that authors ignored the precorneal losses and the barrier function of the cornea. Literature for ophthalmic administration reported that for effective systemic delivery a relatively high drug concentration needs to be circulating in the blood plasma in order to achieve a therapeutically effective dose within the eye.

Response 1: We agree with the Reviewer that the corneal barrier function is important when administering drugs into the eye. It is indeed a major problem to get drugs into the eye due to the corneal barrier, and to improve eye penetration many pharmaceutical ingredients have been used (e.g. surfactants, chelating agents, and penetration enhancers). However, the aim of the present study was to exploit the corneal barriers in achieving high systemic (plasma) naloxone concentrations, and we could show that therapeutic naloxone plasma concentrations can be achieved in the dog via the ocular administration route (i.e. the eye as the administration route and not as a target organ). Therefore, naloxone concentrations in the eye were not measured.

Point 2: The introduction section needs more information about the ocular drug delivery especially the obstacles encountered during ophthalmic administration.

Response 2: We agree that it is reasonable to add more information about the obstacles and the special characteristics of using the eye as an administation route. The Introduction section has now been modified accordingly (Introduction, paragraph 4).

Point 3: Lines 188-192 need rephrasing to make it clear for the reader.

Response 3: As suggested by the Reviewer, we clarified this section (section 4.1, the second paragraph).

Point 4: The conclusion section needs to be re-written. Authors should comments on their finding and the advantages of the obtained results.

Response 4: As suggested, the Conclusions section was modified.

Reviewer 2 Report

The submitted manuscript suggests that eye drops may provide a fast and effective non-invasive method to administer naloxone for reversing opioid overdosing. Eye drops are quick and easy to administer, and they did not cause ocular discomfort in the dog.  the paper is well written and obtained results are interesting for a wide audience of readers, nevertheless, the results presented in Fig. 1 and Fig. 2 are of rather poor technical quality and should be substantially improved using some graphical software.

We also recommend enlarging the References with a paper about naloxone delivery using microneedles:

Papich, M. G., & Narayan, R. J. (2022). Naloxone and nalmefene absorption delivered by hollow microneedles compared to intramuscular injection. Drug Delivery and Translational Research, 12(2), 376-383. doi:10.1007/s13346-021-01096-0

Author Response

Response to Reviewer 2 Comments

Point 1: the results presented in Fig. 1 and Fig. 2 are of rather poor technical quality and should be substantially improved using some graphical software.

Response 1: As suggested, Figures 1 and 2 were redrawn to add clarity, and the resolution was improved. The new Figures were drawn with a GraphPad Prism software, and are provided with the TIF format.

Point 2: We also recommend enlarging the References with a paper about naloxone delivery using microneedles:

Papich, M. G., & Narayan, R. J. (2022). Naloxone and nalmefene absorption delivered by hollow microneedles compared to intramuscular injection. Drug Delivery and Translational Research, 12(2), 376-383. doi:10.1007/s13346-021-01096-0

Response 2: Thank you for informing us about this interesting paper by Papich and Narayan. We added this reference (ref. #17; Discussion, paragraph 4).

Reviewer 3 Report

The manuscript provides valuable data on naloxone systemic delivery via ocular route from a thoughtful experimental design. I have two minor comments.

1. Provide the osmolality of the eyedrop formulations.

2. Provide the statistical levels of significance in tables and figures.

3. How did you achieve dose precision for ocular route @100 micrograms/kg? How did you ensure no spillage of the eyedrop delivered to the ocular surface? In Table 1 and 2, please also provide the volume of ocular solution administered beside the dose.

Author Response

Point 1: Provide the osmolality of the eyedrop formulations.

Response 1: To minimize irritation of the eyes, ophthalmic solutions should ideally have an osmolality of about 300 mOsm/kg, which is the tonicity of the normal tears. However, the eye can tolerate solutions with an osmolality in the range of 200–600 mOsm/kg. We did not measure the osmolarity of the formulations used for these studies, since we did not consider this to be relevant for the study. However, we are aware that the citrate buffer as such is slightly hypotonic (~120 mOsm/kg) and that the formulations containing propylene glycol are hypertonic. Since we did not observe any irritation in the dogs eyes we assumed that the possible hypo/hyper osmolarity does not have major influence  on the current results.

Point 2: Provide the statistical levels of significance in tables and figures.

Response 2: Statistical levels of significance are now provided in Table 2 and in Supplementary Figure. We state in Results (2.1, paragraph 1) that “the plasma exposure of naloxone increased dose-dependently" and in Discussion (the first paragraph) that “Ocular administration resulted in a dose-dependent increase in the plasma naloxone concentration when the naloxone concentration in the eye drop was increased from 3 mg/mL to 15 mg/mL”. Since dose-dependensy instead of dose-linearity or dose-proportionality was claimed, we considered that an analysis of dose-linearity seems unneccessary. Therefore, we prefer not to include statistical significancy testing in Table 1.

Point 3: How did you achieve dose precision for ocular route @100 micrograms/kg? How did you ensure no spillage of the eyedrop delivered to the ocular surface? In Table 1 and 2, please also provide the volume of ocular solution administered beside the dose.

Response 3: We thank for this excellent comment. As response, the headline of Chapter 4.5 was changed to “Drug administration and blood sampling”, and we now provide a description of the drug administration method in the first paragraph of 4.5. The administered volumes of the ocular solutions (in µL/kg) were added to Table 1 and Table 2.